# Risk Assessment for UAS Logistic Delivery under UAS Traffic Management Environment

**Pei-Chi Shao**

Department of Aviation & Maritime Transportation Management, Chang Jung Christian University, Tainan 711301, Taiwan; pcshao@mail.cjcu.edu.tw

**Abstract:** Resulting from a mature accomplishment of the unmanned aircraft system (UAS), it is feasible to be adopted into logistic delivery services. The supporting technologies should be identified and examined, accompanying with the risk assessment. This paper surveys the risk assessment studies for UAVs. The expected level of safety (ELS) analysis is a key factor to safety concerns. By introducing the UTM infrastructure, the UAS implementation can be monitored. From the NASA technical capability level (TCL), UAV in beyond visual line of sight (BVLOS) flights would need certain verifications. Two UAS logistic delivery case studies are tested to assert the UAS services. To examine the ELS to ground risk and air risk, the case studies result in acceptable data to support the UAS logistic delivery with adequate path planning in the remote and suburban areas in Taiwan.

**Keywords:** UAS safety; risk assessment; expected level of safety (ELS); UAV logistic delivery

## 1. Introduction

The advances of micro electro-mechanical systems (MEMS) have led the unmanned aircraft system (UAS) into rapid growth since 2012. The vertical take-off and landing (VTOL) multi-rotor unmanned aerial vehicle (UAV) has made very successful progress in characteristics of simple structure, easy operation, and good performance to suit for wide applications. Thus, UAVs can easily be used to work with dirty, dangerous, and dull (3D) jobs with higher operational efficiency and personnel safety. VTOL UAVs have also successfully grasped consumer markets, and have replaced many conventional systems into new visions.

The International Civil Aviation Organization (ICAO) document 328 for unmanned aircraft systems (UASs) gives the definition of UAS as "*An aircraft and its associated elements which are operated with no pilot on-board.*" VTOL UAVs use brushless (BL) motors, electronic speed controllers (ESC), Li polymer (LiPo) battery, GPS navigation, and inertia navigation microcontroller to build on fiber materials with a required payload through the radio link to a ground controller to fly. This creates a friendly UAS operation environment [1,2].

The UAV flight operation and management in Taiwan is legislated by the Ministry of Transportation and Communications (MOTC), to approve the legal use of UAVs in non-integrated airspace (NIA) by the "Civil Aviation Act". This was legislated on 3 April 2018 [3] and is effective on 31 March 2020. This UAV law regulates UAS operations and is enforced into surveillance and management to assure the UAS flight safety. Research and development on the UAS traffic management (UTM) with appropriate UAS business models and applications are booming. Several UAS services and business models are established for journalist/news broadcasting air photography, traffic surveillance, agriculture insecticide spray, mountainside slide inspection, bridge inspection, and logistic delivery, etc.

The Federal Aviation Administration (FAA) advisory circular (AC) no. 107-2 states that the operational limitations for small UAV (sUAV) are to fly less than a ground speed of 160 km/h and lower than 400 feet above ground level (AGL) [4]. AC 107-2 by FAA allows all the applications of sUAV

to be legitimated for delivery of goods, surveillance, and search and rescue by Kopardekar et al. [5]. Figure 1 shows a typical commercial legal application involving Amazon's logistics trials using sUAVs in NIA for air traffic control (ATC) under 400 feet AGL [6,7]. The European Union Aviation Safety Agency (EASA) and Single European Sky ATM Research (SESAR) also define the U-space concept for sUAVs NIA below 700 feet [8,9]. Different regulations from FAA and EASA can be adopted to suit for UAV operations to different countries and territories.

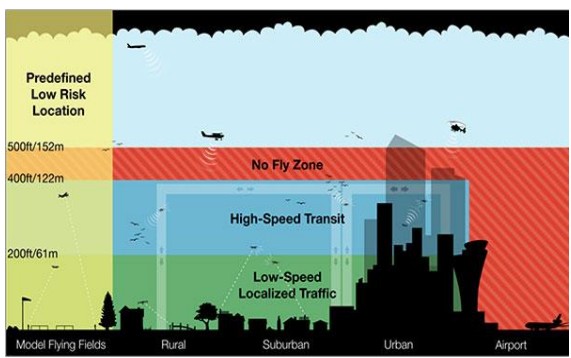

**Figure 1.** Amazon's non-integrated airspace for unmanned aerial vehicles (UAVs) [6].

In Taiwan, the NIA below 400 feet is authorized and commissioned to local governments for management; while higher altitude flights are controlled by the CAA authority and air traffic control (ATC) [3]. Under such circumstance, a hierarchical UTM is designed and constructed to include all UAVs into regional UTM (RUTM) or national UTM (NUTM) [10,11]. The hierarchical UTM proposes an ADS-B like infrastructure with an on-board unit (OBU) to broadcast flight data down to ground transceiver stations (GTS) to UTM cloud for surveillance. The surveillance data include position track and flight data with six-degrees-of-freedom, to transparently watch all UAVs flying in the responsible airspace. The preliminary tests verify the importance of introducing UTM for UAV surveillance [10].

Due to the lack of statistical data for the past UAS flight experience, it is difficult to collect real flight data for risk analysis. From the literature survey, many studies adopted general aviation (GA) data to simulate UAV risk assessment [12]. However, UAVs are indeed increasingly used for commercial or private activities, further studies in deep involvement become urgent and critical. The developing UTM system tries to collect UAV big data from UAV surveillance with flight data for flight operation quality assurance (FOQA) analysis [10]. UAS FOQA can support UAV risk analysis with real data. The safe operation of sUAVs becomes a great concern to the public and turns into a significant challenge to aviation safety. Studies include risk assessment concerning airspace, UAV MRO (maintenance, repair, and overhaul) in manipulation [1,13]. To investigate the UAS risk and safety, a logistic delivery integrated pilot program (IPP) is demonstrated for technical verification in this paper.

On the UAV risk assessment, it is termed into air risk and ground risk classifications [14,15]. In the UAV air risk, collision avoidance resulting from geofensing, detect and avoid (DAA) mechanism, and front detect is focused on. While in the UAV ground risk, crash mode [11], ground impact, kinetic energy, and debris casualty [16,17] due to failures were analyzed in profundity and have been paid highly attentions in the UAS operation.

A typical risk assessment study focusing on sUAVs was undertaken by the FAA [1,12] to determine the risk level of flying sUAVs over different types of areas. In this report, only small UAVs of less than 250 g and flying at less than 25 m/s were taken into account. However, in reality, a velocity of 25 m/s can only be achieved in fixed wings [1]. From real experience, multi rotor UAVs are flying 4~8 m/s with their maximum take-off weight (MTOW) around 10 kg that may be a typical carrier for delivery. In addition, wind effect may interrupt sUAS flight operations due to the small flight momentum. The flight conditions need verifications by real flights. Logistic delivery flight cases will be discussed in this paper.

This paper accounts an UAS risk assessment study using flight scenarios that involves the FAA formulation for commercial UAVs of 12 kg MTOW flying at 5–8 m/s [11]. The logistic delivery case study is used to support the analysis of risk assessment and the expected level of safety (ELS) and for further estimation on risk level to population density in remote and suburban areas.

UAV accidents are monitored by the aviation authority and police administrations. The Civil Aviation Act legislates regulations for UAV flights. UAVs can plan routine flights in legal airspaces, which are colored as yellow and green areas, except the restricted red areas in Taiwan [3]. The operation of UAVs brings a significant facilitation of convenience but impacts public safety.

Before 2004, without adequate regulations, UAVs were governed by the following guidelines [13]:

*UAV Operations Shall not Increase the Risk to Other Airspace Users or Third Parties*

The Joint Aviation Authorities (JAA) used an "Equivalent Risk" for UAV operations [13] to monitor the safety of UAVs and UAV accidents for the past 15 years. Due to the lack of UAV event/accident data, manned aircraft event/accident data have been referred to determine the risk level [13,18] for UAVs. In order to determine the risk of UAV operations, some safety factors are proposed using the linear model, such as the physical factors of weight, velocity, kinetic energy (KE) and frontal impact area [11], the ground population, and the effect of shelter and the number of casualties [18,19]. Frequent UAV activities in the near future imply that air traffic monitor and control in the low-altitude airspace is required with an effective methodology in surveillance [5,10].

"Specific Operations Risk Assessment" (SORA) by the Joint Authorities for rulemaking of unmanned system (JARUS) [15] claims that the UAS risk is a combination of probability of any associated levels of severity in occurrence. The safety level defined by probability fatalities is classified on the ground or in the air. The SORA provides a systematic methodology to identify risks associated with an UAS operation in a holistic way. The SORA process is a valuable tool to standardize the risk framework in UAS operations. For example, in order to reduce the air risk for mid-air collision, the SORA takes the tactical mitigations by the DAA mechanism or alternate means of services of operational procedures. Since DAA is one of the ways to mitigate the air risk in mid-air collision, the hierarchical UTM system can detect UAVs in the approach and command and avoidance by software manipulation. With real data performance through real time flight surveillance, software DAA is achieved [10]. A principal task of UTM is applied to accurately determine the safety level of UAVs.

An UAV delivery is demonstrated as an integrated pilot program (IPP) for UAS under UTM surveillance. It is used to verify the feasible use of UAS in logistic delivery. From the results, it does strongly enable the risk level assessment from such UAS applications.

A hexa-rotor VTOL UAV is tested in a remote area and suburban area for risk assessment under UTM surveillance. The purpose of the tests is to verify the technical feasibility and capability. The quantitative analysis is not focused on in this phase of experiments. The flights carried 2.4 kg flying at 35 m above ground level (AGL) for delivery over 5 km away. The demonstrations were successful in terms of autopilot performance flying beyond the visual line of sight (BVLOS). To determine the safety use of UAV in logistic delivery, the risk analysis for this case study has arisen to take flight planning into account [11]. Although the mean time between failures (MTBF) for VTOL UAVs is still as low as 100 h, the risk assessment for this study results in a safety measure to banish society concerns to UAS operations.

## 2. Risk Orientation

The UAV risk assessment in terms of safety management uses the pilot experiences and safety records for manned aircraft [12]. However, UAVs carry no passengers, so the safety standard for UAV can be focused on the protection of third parties and property. To increase public safety, an equivalent level of safety (ELOS) for manned aircraft was used to certify UASs by the Joint Aviation Authority (JAA) in 2004 [12,20]. Weibel and Hansman [17] demonstrated the use of ELOS to determine the operational

requirements for different classes. However, it is difficult to quantify an ELOS for UAV applications due to the lack of real data. Dalamagkidis et al. used the standard components of the system to establish a target level of safety (TLS) [21]. Neither ELOS nor TLS can be used for a real UAV safe assessment using the simulated data. An equivalent analysis concept is used to determine the similarities in risk development for UAVs and manned aircraft.

UAV technologies are elevating into maturity rapidly. Runaway pilots operating incontrollable UAVs have threatened aviation safety near airports in Taiwan since 2015 [3]. An orderly regulated UAV surveillance under UTM is expected. Recent studies of UAV risk relate to the sUAV operational safety in low-altitude airspace focusing on UTM [22], ground impact hazard [17], the third party casualty risk [1,13], and the target level of safety (TLS) [18,21]. Under scheduled IPP flight tests, the UAS operation safety or risk can be analyzed and estimated from the observable and controllable data.

*2.1. Ground Risk Assessment*

The SORA [16] describes the guidelines to approach safety created by the UAS operations for specific assurance and integrity levels (SAIL) into either ground risk and air risk. Both risk and SAIL can be reduced by the effective methodology. It can be accomplished through the UAS operators by utilizing certain threat barriers and mitigating measures. From which, UTM is one of legal, feasible, and effective solutions. Under such understanding, risk assessment and prevention from ground and air shall be taken into account seriously. Most studies refer to the general aviation or air transport aircraft. However, this is indeed impractical. In this study, the risk assessment study considers some preliminary tests with certain scenarios to assert a feasible evaluation.

The event tree analysis was used to analyze four scenarios for harm to the public on the ground from the effect of UAV operations to impact public safety. The events include: (1) Failure of UAS, (2) impact in the populated area, (3) debris penetration to sheltering, and (4) resulting fatal penetration [13,16,17]. Lin and Shao [10] analyzed the UAV air crash behavior to explore the severity of ground impact resulting from the experiments. The UAV ground impact event tree shows the risk factors for UAV failure. The "Ground Impact Hazard" is a model to determine the effect of different factors on the expected level of safety (ELS) [17,23,24]. It is expressed in terms of ground fatality events per hour of flight (E/h). The factors for ground impact include the total system reliability, UAV size, UAV kinetic energy (KE) at the moment of power loss, and population density near UAV flight operations [17]. Since there is very little data about UAV fatalities [15], the TLS for the ground impact model uses a value of $1 \times 10^{-7}$ E/h, which is recommended by the FAA for manned aircraft operations. In the reference, the TLS for air transportation ground fatalities is $2 \times 10^{-7}$ E/h, which is recommended by the National Transportation Safety Board (NTSB) database. These data can fit either manned aircraft or unmanned vehicles. According to these data, the ELS for the ground impact model for an UAV is $1 \times 10^{-8}$ E/h fatalities [17].

Melnyk et al. [13] used a linear model to estimate casualties and determine the UAS risk. The model parameters include area population, shelter effect, and frontal impact area casualties. In terms of the linear model, the parameters for UAVs are frontal impact area [10], kinetic energy, and the effect of shelter [1,20].

There is a wide range of UAS sizes and characteristics, such as the on-board flight control system and/or the presence of a communication link. Therefore, the model for manned aircraft ELOS cannot directly apply to UAVs because an UAV does not carry passengers and crew. The probability of injuries and fatalities for UAVs are lower than that of any kind of manned aircraft [21]. Since the various accident types have resulted in different effects on the safety level, the ELOS, TLS, and ELS must be categorized for UAV operations of different types, power, weight, and altitudes using similar accident scenarios for risk analysis. This is an important key to UAS. An UAS database must be established to assert risk assessment from the causes of failure to reduce the estimation error. A preliminary study on the construction of an UTM by Lin et al. [10] acquired flight control data on the UAV performance for flight operation quality assurance (FOQA).

The SORA methodology by JARUS [16] provides the ground risk class (GRC) determination to identify the ground risk level using the risk scores of UAS categorizations of operation. The ground risk level determined by the (1) maximum UAS characteristics dimension from 1, 3, 8 m or larger, (2) their corresponding typical kinetic energy from 700 J, 34 kJ, 1084 kJ or larger, and (3) operational scenarios, such as VLOS, BVLOS, and population area, their intrinsic UAS ground risk classes vary from low to high. Each different condition of class impacts the severity of ground risk. It is obvious that the UAS kinetic energy causes a dominant casualty to the ground, either to human life or property.

Lin and Shao [11] used the Weibel's ground impact model [17] to define possible injuries and fatalities, and ELS is the risk assessment tool to identify the risk level. The simulation resulted from an effective path planning for least crash probability density (CPD) [11].

To determine the damage due to the UAV failure, the physical characteristics of the UAV are used from the FAA in experiments. The most significant factors involve UAV weight and kinetic energy [1,10]. The FAA task force used information from a study by the United Kingdom Ministry of Defense in 2010, which states that an object with a kinetic energy level of 80 J has a 30% probability of striking the head of a person [1,5]. In terms of the number of casualties for a threshold value for mass and velocity, this equates to an object weighing 250 g traveling at a terminal velocity of 25 m/s (approximately 57 m/h). Although the terminal velocity of 25 m/s used in this report is not realistic in experience, the impact harm and injury to people on the ground resulted from failures.

In order to determine the probability of a catastrophic event involving an UAV, the FAA task force uses the mean time between failures (MTBF), the population density and the exposed fraction and probability lethality to calculate the probability of an UAV event that results in casualties. The results of experiments by the task force mean that FAA does not require the registration of any UAV with a MTOW < 250 g (0.55 pounds) [4].

Based on the UAV flight path risk, Lin and Shao [11] developed the crash probability density (CPD) radius of UAV path planning. The ELS can provide risk level information to avoid the high population along the path planning in the pre-flight process.

In medium- and high-risk environments, UAS applications for logistic delivery over populated areas are allowed in the near future. The European Union Aviation Safety Agency (EASA) releases a special condition to certify UAVs for logistic services. It is open to the public for comments before 30 September 2020 [25]. The proposed certification approach, SC-light UAS, will apply to all UAVs with a MTOW < 600 k (1322 pounds). However, this does not allow transporting passengers in any way, which is operated without a remote pilot being able to intervene. The document also applies to UAV operations in terms of "specific". The EUs risk-based framework is defined as open, specific, and certified. Rulemaking on the certified category is on-going [25]. In the near future, the urban air mobility (UAM) vehicles may become a different story to discuss.

## 2.2. Air Risk Assessment

The NASA UTM technical capability level (TCL) is surveyed based on the four metrics about ground and air risks with UAS operations: (1) Population density, (2) the amount of people and property on the ground, (3) the number of manned aircraft in close proximity to the sUAS operations, and (4) the density of the UAS operations [26]. The TCL verifies and confirms the maturity of UAVs in urban and suburban services, especially logistic delivery.

For the air risk management in NAS airspace, Melnyk et al. [18] proposed a framework to develop an effectiveness standard of sense and avoid (SAA) for UAS. "Effectiveness" is defined as the combination of reliability and efficacy, which also indicates UAS failures or insufficient performance standards. In order to develop the minimum effectiveness standard for SAA, a framework is utilized to include a target level of safety (TLS) approach to the problem and an event tree format risk model to predict mid-air collision (MAC) fatality rates resulting from UAS operations. The event tree model is a risk mitigation by proper separation or collision avoidance. The event tree is a series of branches

with options for the air environment, mitigation, and event outcomes based on the probability of each of the branches in occurring and the effects of progress along a branch [18].

Since UAS needs a vehicle function certification to ensure or minimize with the least in-flight collision and ground impact fatality, this is defined as target levels of safety (TLS). Therefore, Schrage developed the sUAS operators that need a functional safety management (FSM) approach that is affordable to ensure safety for their limited operations. The air risks are including the functions or subsystems of UAS failure. The risk assessment tools for functional hazard assessment (FHA) and operational risk assessment (ORA) are utilized to complete the UAS safety assessment process in the UAS logistic delivery experiment [27]. FHA identifies and evaluates the hazards associated with functions of the operation system, while ORA evaluates the overall risks associated with each hazard of function. TLS is used as a constraint for sUAS functional safety management (FSM). The key purpose of FHA identifies and evaluates the hazards associated with aircraft-level functions. Based on the research results, the functions associated with failures will have the highest risk dealing with flight control and SAA capabilities. The functional decompositions of sUAS for logistic delivery referring to different risk levels are constructed in block diagrams. From this block diagram, SAA is the most significant area for risk mitigation as important as guidance, navigation, and control (GNC) in the flight control [27].

Martin et al. addressed [28] that SORA adopts a holistic view for managing air risk, incorporating greater flexibility. This indicates how mitigation can be combined with a strategic or tactical way. The flexibility deals with a qualitative set of rules on traffic density with a continuous performance function of detect, decide, and avoid. The SORA employs three air risk classifications (ARC), where ARC-b, c, d are necessary equipment integrity and assurance requirements. ARC is a qualitative classification for the rate at which an UAS would encounter manned aircraft in NAS. It is an initial hypothesis for aggregated collision risk in the airspace, before any mitigations may be applied.

Allouch et al. [14] followed the ISO 12100 to approach risk assessment and risk mitigation by three step risk analyses: (1) First, to start with system limits specification in five categories on physical, temporal, environmental, behavioral limits, and networking limits. (2) Second, performing hazard identification to provide a list of potential drone hazards according to their external and internal sources. (3) Third, estimating UAV risk measures of probabilities and severity levels of the consequences of the identified UAS operational hazards. According to the ISO 12100 standard, the risk estimation consists of determinations on the risk severity and probability. The risk severity is estimated based on the injury level or the harmful impact on people, environment, and UAV itself. The risk severity of the hazard is usually affected by the degrees of consequence as catastrophic, critical, marginal, and negligible; while the risk probability is recognized by frequent, probable, occasional, remote, and improbable occurrence. The hazard sources from internal and external affecting factors are important to UAV pilots to prevent from malfunction and failure.

The 4G/LTE communication is selected as one possible implementation of information exchange by Allouch et al. [14]. The procedures are specified into pre-flight, in-flight, and post-flight. Each phase needs to establish a standard operation procedure (SOP) to assure that the pilot, vehicle, and environment are being ready and suitable to perform an UAS mission flight. In recent research, Lin and Shao [10] demonstrated the ADS-B like infrastructure to establish a surveillance down link into the UTM.

From the literature reviews, UAV operations have different levels of risk including flight procedure, system infrastructure, CNS, meteorological and environmental factor, and human factor. Different types of UAVs with various specifications may result in different levels of risk assessment referring to methodologies and suited regulations. These are also varying in different countries under different national regulations in traffic management under low altitude, with 400 feet by FAA UTM and 700 feet in EASA U-space.

## 3. Integrated Pilot Program for UAV Logistic Delivery

To examine the risk level of UAV flight operations, two cases of logistic delivery will be carried to look into the details of risk analysis. The IPP is a typical example for system performance verification.

### 3.1. UTM Environment for IPP

In the UTM system [10,29], the ADS-B like infrastructure plays an important role to cover 400 feet of regional UAS surveillance. The ADS-B like technology develops ground transceiver station (GTS) and on-board unit (OBU) from 4G/LTE (long term evolution), APRS (automatic packet reporting system), LoRa (long range wide area network), and XBee, for data link to UTM, as shown in Figure 2.

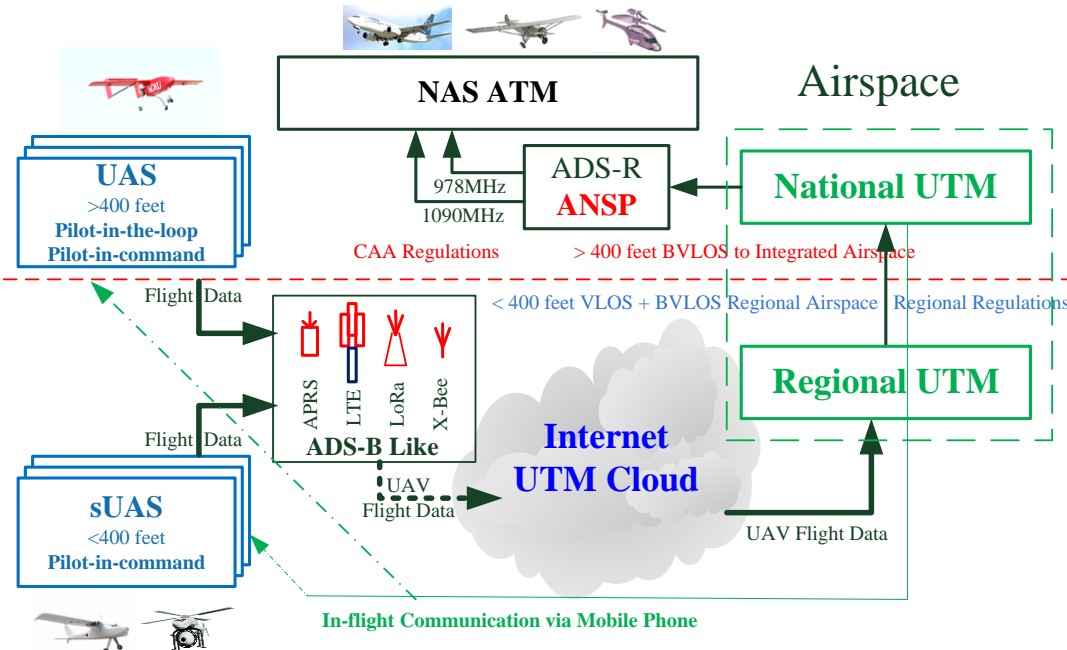

**Figure 2.** The hierarchical unmanned aircraft system (UAS) traffic management (UTM) in Taiwan [10,29].

Similar to the 4G/LTE base transceiver station (BTS) in mobile communication, three other types of the ADS-B like technology need to deploy and build their specific GTSs for territorial coverage to relay UAV surveillance data into the Internet to the UTM cloud. The first region UTM was constructed in Tainan City with five LoRa stations, as shown in Figure 3 [19] for proof of concept (POC). These five GTS sites are CJCU, Yujing, Baihe, Yanshui, and Xigang, as marked in Figure 3. The flight tests have been verified with full surveillance coverage in Tainan City under UTM operation [10,29].

Two types of OBUs are selected for tests in this paper, a 4G/LTE cell phone and a LoRa OBU, as shown in Figure 4 [29], corresponding to its infrastructure.

### 3.2. Processes for IPP

This study tries to examine the UAV flight trajectories to estimate risk prevention functions, such as risk assessment and UTM surveillance. In the operation, the integrated UAV risk prevention can be examined by Figure 5 in real time including air risk and ground risk [14,15,19]. The air risk is focused on UAV collision avoidance via DAA, while the ground risk concerns the human injury due to crash. For sUAS, time to conflict (TTC) among UAVs is conducted by the DAA software in UTM server [30]. The performance is feasible to implement since sUAS is not capable of carrying additional space or payload to carry the detection hardware on the airborne.

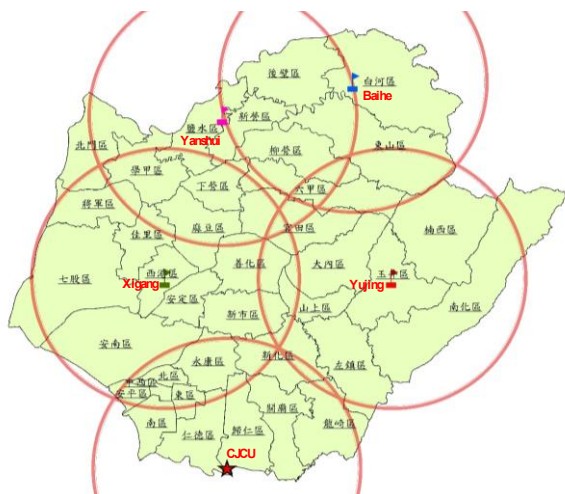

**Figure 3.** Long range wide area network (LoRa) ground transceiver station (GTS) deployment in Tainan for the first proof of concept (POC).

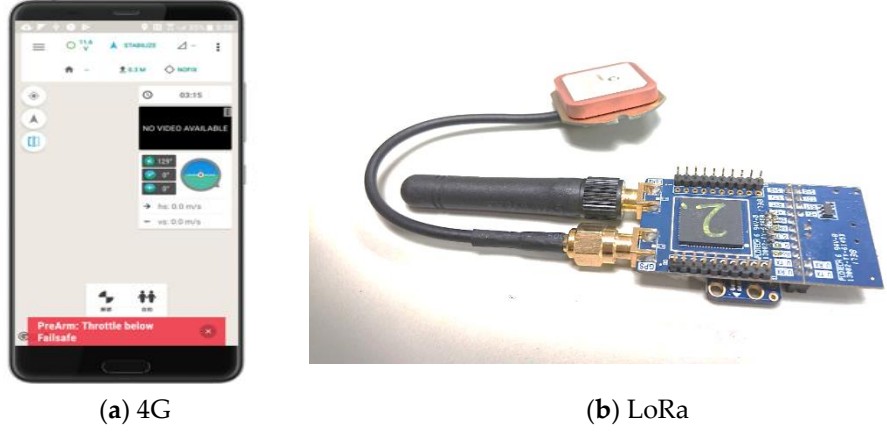

(**a**) 4G                                                                    (**b**) LoRa

**Figure 4.** Automatic dependent surveillance–broadcast (ADS-B) like on-board units (OBUs) for the 4th generation mobile communi cation long term evolution (4G/LTE) and LoRa.

Referring to Allouch et al. [14] of a useful flight process, the UAS provider or operators should set up standard operation procedures (SOP) to carry pre-flight, in-flight and post-flight. In Taiwan, the UTM flight procedures are also with three similar phases. In the pre-flight phase, the CPD path planning [11] and ELS [17] are applied to check referring to the territorial geodetic information to by-pass the densely populated areas. The pre-flight procedure is shown in Figure 6. Pilots and their UAVs are required to log-in from the UAS management information system (MIS) by CAA [3]. The MIS includes databases of licensed pilots, registered UAVs, and no-flight zones (NFZ). In MIS, red, yellow, and green areas are marked to identify restricted (red), conditioned (yellow), and free (green) airspace to fly. The pilots need to submit the flight plan into UTM for approval. UTM will check the proposed flight route to keep away from the red NFZ. Since multi-rotor UAVs can fly no longer than 60 min at 8 m/s velocity at present, the flight route will not be farther than 28 km in its surveillance range. The communication test between controller to pilot should adopt either the 4G/LTE cell phone or Zello broadcast [10,29].

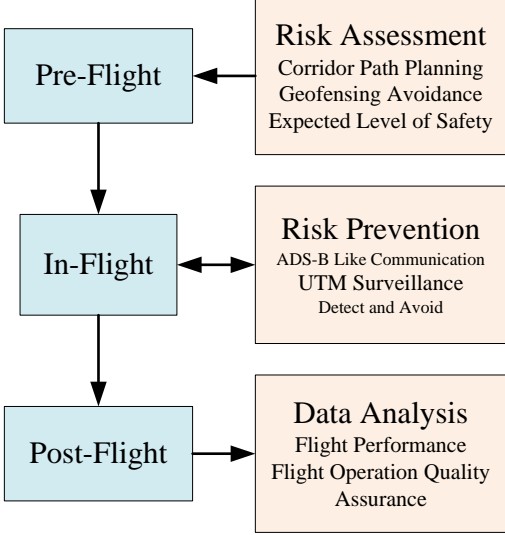

**Figure 5.** The concept of integrated UAV risk prevention system.

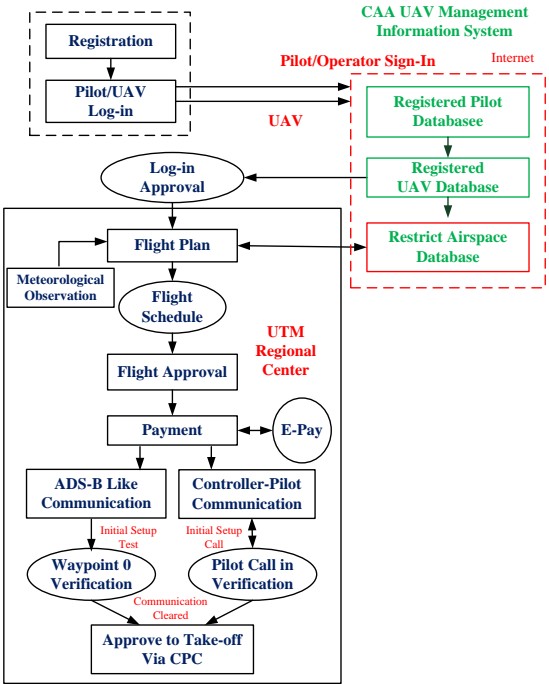

**Figure 6.** UTM pre-flight procedures.

When the flight plan has been approved and the airspace is clear to go, the UTM controller will issue a clearance to the pilot for take-off, as shown in Figure 7. In the in-flight phase, UAVs have been equipped with an ADS-B like OBU for surveillance [10,29]. ADS-B like OBU broadcasts flight data down to GTS and connects into the UTM cloud. The UTM controller will monitor UAV flights and offer flight tracking with no path violation. The DAA performance will be carried on the UTM server software with conflict detection and resolution. Once an UAV separation violates, the UTM controller will intervene by the controller-pilot communication (CPC) for conflict resolution advisory (RA) [10,30]. The UTM software DAA creates a mechanism similar to the separation bubble in TCAS, as shown in Figure 8. When multiple UAVs appear in a small window (range), the DAA will be

activated. UAV speed extrapolations will check the possible time to conflict (TTC) for the next few data intervals. TTC generates warning signals for traffic advisory (TA) and resolution advisory (RA), where TA = 48 s and RA = 25 s. In UTM, the surveillance data interval are regularly set from 8–10 s. Figure 8 shows the concept of DAA referring to TCAS by ICAO. DAA RA is performed by CPC. The priority assessment follows the air traffic control rule.

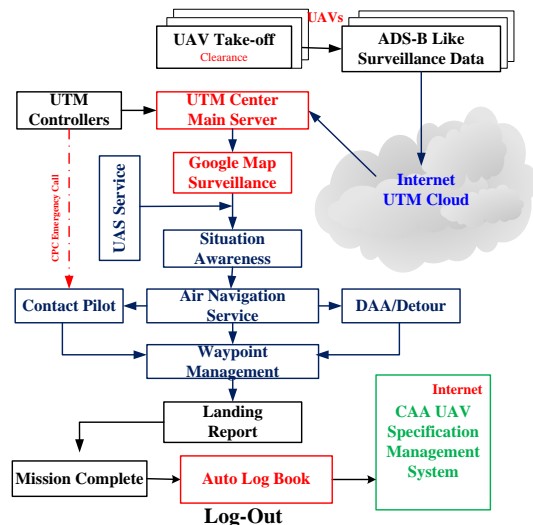

**Figure 7.** UTM in-flight and post-flight procedures.

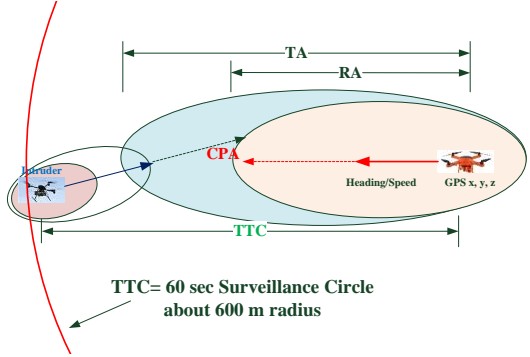

**Figure 8.** Concept of UTM detect and avoid (DAA) mechanism.

The UTM server extrapolates UAV headings to check their separation by TTC. If a conflict is possible in the next few time intervals, an alert will generate to the UTM controller. The UTM controller will check the priority and contact the less priority pilot to detour via CPC. CPC is activated using a cell phone or Zello broadcasting. In Figure 9, AK 1035 has less priority to detour by a right turn of 15 degrees to avoid.

In the post-flight phase, the UAV pilots need to log out from the UTM and CAA MIS. The ADS-B like reports 90 byte data including pilot ID, UAV ID, GPS position, and six-DoF flight data. It appears as follows:

[Heading(5); UAV(6); Pilot(6); Lat.(9); Long.(10); Alt.(4); 6 DoF($p, q, r, \alpha, \beta, \gamma,$) (36); V(6); A(6); Tail(2)]. The short one is only UAV ID, Pilot ID, and X, Y, Z data. After flights, the flight performance can be analyzed for flight operation quality assurance (FOQA) using the UTM surveillance data.

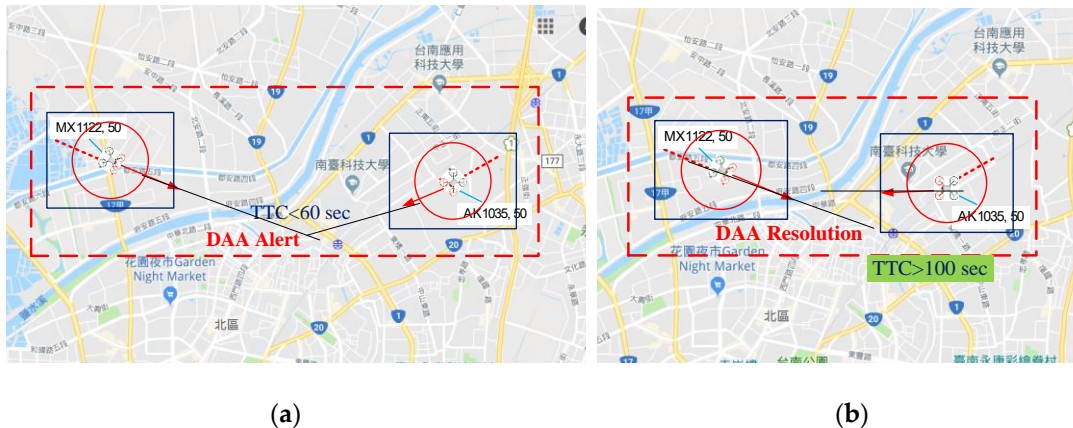

(**a**)　　　　　　　　　　　　　　　　　　　　　　　(**b**)

**Figure 9.** Conflict alert and resolution, less priority UAV avoids (**a**) TTC < 60sec with DAA alert; (**b**) TTC > 100 s with DAA resolution.

### 3.3. UAV Delivery Case Study

This study concerns the use of UAV for logistic delivery in case studies. The flight system and test specifications are described in detail. The purpose of the delivery case tries to examine the effectiveness of surveillance capability of ADS-B like infrastructure under UTM, and further to analyze UAV expected levels of safety (ELS). Two cases are demonstrated for the logistic delivery scenario.

An hexa-rotor UAV is used to deliver a parcel with a weight of 2.4 kg (four bottles of water). The UAV flies at a velocity of 5–8 m/s and cruises at 35–50 m AGL for less than 40 min. The flight performance adopts a Pixhawk flight control autopilot with a Google map mission planner for beyond visual line of sight (BVLOS). The flight is fully monitored by UTM using ADS-B like technology.

The first flight case is flying in Ping Tung County. UAV carries the 4G/LTE cell phone for flight operation and video surveillance. A 900 MHz communication is added for the control uplink. A QR code is placed on the ground as the final target. The video downlink aims at the target with image processing in the QR code recognition to accurately locate the delivery target via 4G/LTE. This scenario performs the UAS delivery over a river, where ground transportation detours a long router. The surveillance uses 4G/LTE to report the flight track via the Internet to UTM.

In the second flight case, the UAV delivery is from CJCU to Hsin-Ta Harbor. The delivery goes directly from CJCU to the destination. However, ground transportation needs to detour with several junctions. The flight is monitored under the Tainan RUTM. The UAV flight data are collected and broadcast into ground transceiver station (GTS) via the Internet to the UTM cloud. This test uses the LoRa ADS-B like on-board unit (OBU) for real time cloud surveillance [10,19].

### 3.4. Risk Mitigation from Path Planning

In the pre-flight phase in Figure 6, the first flight case was operated in a remote area, where the density of population results in less safety concerns. In the first scenario, Figure 10 shows delivery across a river in Santiman, Pingtung County. The UAV delivers a small parcel of 2.4 kg to the ultra-light field at the other side of river, which is about 1.6 km away. Since the area is a countryside, the route is planned point to point and is flown at a velocity of 5–8 m/s. The test was conducted on a clear sunny day, wind miles/h from azimuth 320. The UAV flew a fair wind to cross the river. The actual flight time was about 8 min and the UAV flew at a constant altitude of 35 m above ground level (AGL). The 8-min UAV flight would require 25 min for ground transportation. In this flight test, 4G/LTE of the selected ADS-B like is adopted for surveillance into the UTM. This test just tries to verify 4G/LTE as a choice of ADS-B like technology wherever the BTS can cover. Figure 10a shows the path planning and 5b shows the real flight surveillance.

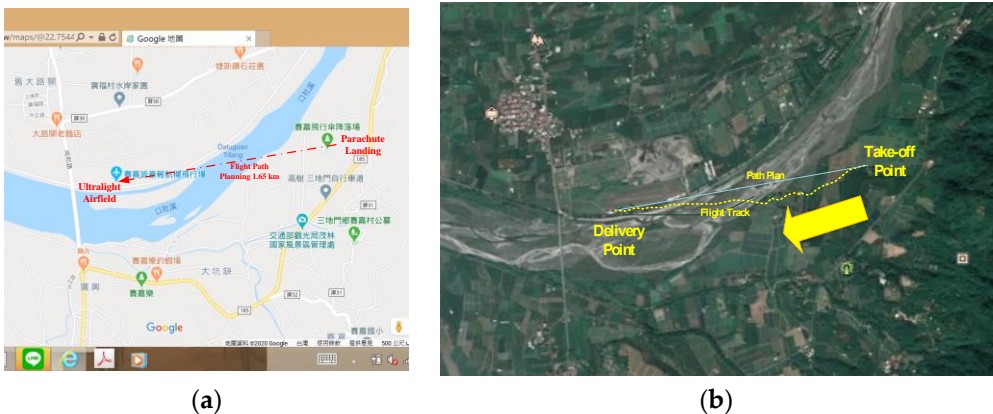

**Figure 10.** UAV delivery test 1 at Santiman across the river under 4G to UTM (**a**) Test 1 path planning; (**b**) Test 1 UTM tracking.

In the second scenario, the delivery takes place in a suburban area at Queiren, Tainan. The UAV delivered a parcel from CJCU (Chang Jung Christian University) to Hsin-Ta Harbor, which has a direct distance of about 8 km, as shown in Figure 11. Figure 11a shows the path planning with GTS coverage under Tainan RUTM. The test uses LoRa OBU to report UAV surveillance data to LoRa GTS. In Figure 11a, the CJCU GTS has its coverage of 15 km. The arrows show the range within the GTS coverage. Test 2 was flown on a clear sunny day with wind 3 m at azimuth 020. It is a fair wind for flight. A geo-fence mechanism was activated to keep the UAV away from the restricted area of the airport. This test flies over the freeway. On the delivery path planning, the UAV has an altitude trajectory constraint to pass the freeway by 50 m above. The total flight time was 24.5 min, cruising at 50 m AGL. For test 2, the UAV took off at 35 m elevation and landed at 5 m elevation at Hsin-Ta Harbor. In this flight scenario, it takes 67 min on the ground traffic. Since the second scenario is performed near the airport, the UTM controller will monitor the UAV logistic operation from intrusion into the restrict area.

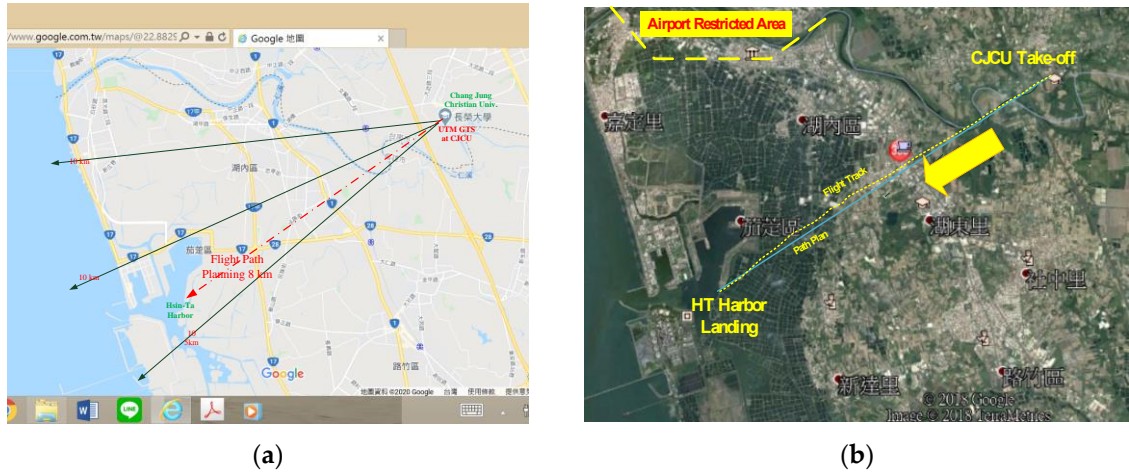

**Figure 11.** UAV delivery test 2 from CJCU, Queiren to Hsin-Ta Harbor under UTM (**a**) Test 2 path planning; (**b**) Test 2 UTM tracking.

## 4. Risk Assessment for UAS Delivery

From the logistic delivery tests, the selected territories are either remote or suburban with less population. This is the most feasible condition for UAS logistic delivery at present. How is the risk level to adopt UAV into logistic delivery? Using FAA simulations, this paper examines the expected level of safety (ELS) in these two demonstrations.

*4.1. FAA Risk Level*

The risk assessment is given in terms of the FAAs report in 2015. The risk level is evaluated by the probability of events. A sUAV (<25 kg) failed to free-fall to the ground [1]. The FAA scenario applies to the UAV flying above a certain population density ($n/m^2$) concerning the MTBF for the specific UAV, the frontal impact area of the UAV ($S_{UAV}$), the impact area of humans ($S_h$), the kinetic energy of the sUAV (KE), the exposed fraction (*EF*) of humans, and the probability of lethality ($P_l$) for impact casualties for an UAV with MTOW (*M*) and flight velocity (*V*). The population density is calculated by the total ground area of the surface ($S_S$) to the number of humans (*n*).

The KE of the sUAV is determined using the terminal velocity of the sUAV, the MTOW (*M*), and the drag coefficient ($C_d = 0.3$), as:

$$\text{KE} = \frac{1}{2}MV^2 \tag{1}$$

The FAAs risk level is calculated using the probability of a sUAV event as:

$$P_{event} = \frac{S_{UAS} \times \left(\frac{n}{S_s}\right) \times EF \times P_l}{MTBF} \tag{2}$$

where:

$$\text{Population Density} = \frac{n}{S_s} \tag{3}$$

In the pre-flight phase of Figure 6, this paper uses the assessment formula and modifies the assumptions for the real situation to calculate the risk level for the delivery scenario. Two flight test results at Santiman, Pingtung and Queiren, Tainan were used to estimate the safety level of delivery using the population density figures for 2018 for suburban and remote areas from the Ministry of Interior (MOI), Taiwan [31]. The risk assessment data are listed as follows:

a.  Selected UAV: Arm pitch 83 cm hexa-rotor, MTOW *M* = 12 kg, cruise speed *V* = 8 m/s.
b.  Population: $n/S_S$ = 0.00013 (Santiman/remote area, 130 $n/km^2$) and 0.000651 (Queiren/suburban area, 651 $n/km^2$).
c.  MTBF: 100 h, in accordance with the FAA data [1,4].
d.  Area of UAV ($S_{UAV}$): 0.6889 $m^2$, arm pitch 0.83 × 0.83 m from a hexa-rotor.
e.  Exposed Fraction (*EF*): 0.2, in accordance with the FAA data [1,4].
f.  Probability of lethality ($P_l$): 0.3, in accordance with the FAA data [1,4].
g.  Kinetic energy of UAV (KE): 384 Joules, from a hexa-rotor.
h.  MTOW (*M*): 12 kg.
i.  Velocity (*V*): 8 m/s (maximum operating speed of hexa-rotor H83).

The results are shown in Table 1. The most significant differences are seen in the KE, *M*, and *V* terms. Referring to the FAA simulation uses 250 g at a speed of 25 m/s in an area with a high population (10,000 $n/m^2$ or 3853 $n/km^2$), the flight tests in this paper use the hexa-rotor H83 flying with 12 kg at 8 m/s speed and at an altitude of 35 m. In terms of the population density, the probability of an event in a remote area (Santiman, PT) is less than that for a suburban area (Queiren, TN) [31]. In comparison, the respective risk level for a commercial air transport and general transport is $1 \times 10^{-9}$ E/h and $5 \times 10^{-5}$ E/h, and the risk levels of this study, $5.37 \times 10^{-8}$ and $2.69 \times 10^{-7}$, in Table 1 are reasonable. According to the FAAs statistical data for general aviation (GA) fatal accident rates from 2010 to 2017 [12], the average GA fatal accident rate is $1.028 \times 10^{-5}$ per 100,000 flight hours. The probability for an UAV that is calculated by this study is less than the GA fatal accident rate [12]. The lack of safety data for UAV has no direct effect on the reliability and safety level compared with the GA and commercial air transport safety records. Experience shows that due to the gyroscopic effect of the multi-rotor system [11], the quad-rotor does fall in a spiral trajectory. The impact to the ground is different from that of free-fall. Table 1 shows that the terminal velocity of the sUAS is 25.7 m/s [1,11].

This is an exaggerated result in terms of the real performance for a sUAS. This study uses a maximum terminal velocity of 8 m/s [11] from the actual flight experience.

The UAVs weight and speed (KE), the population density, and the failure rate have the greatest impact on the number of casualties [13]. The results of the risk assessment for this study show that future research should focus on the reasons for the failure of an UAV. UAV performance parameters, such as MTOW and velocity, determine the severity of an UAV failure for a specific population density, so the reliability of an UAS must be improved in terms of the *MTBF*. Further study of risk prevention and risk management for UAS services is necessary to increase the safety to the public.

### 4.2. Analysis on Expected Level of Safety (ELS)

The level of safety is calculated for the purpose of integrating different types of UAV into the NAS regarding the safety requirements. Based on Weibel's ground impact model [21] in Figure 12, the possible injury and fatality are taken into account. Moreover, the expected level of safety (ELS) is calculated using Equation (4). In Figure 12, the air risk model is also descriptive to the failure of DAA. Two UAVs might collide to crash to turn into this ground impact model. It is also important that the UTM controller is highly responsible to pay attention to the DAA alert when multiple UAVs appear in the same area. In Figure 8, the UAV icon on the UTM display can extrapolate by their heading arrows by 5 to 10 times of the surveillance data period of 5–8 s. It will examine the TTC of two UAVs by their TAs and following RAs.

$$ELS = \frac{1}{MTBF} A_{exp} \rho_p P_{pen} (1 - P_{mit}) \tag{4}$$

*ELS*: Expected level of safety (failure/hour).
*MTBF*: Mean time between failure of UAV.
$A_{exp}$: Area of exposure (m$^2$), or frontal impact area (FA) of UAV.
$\rho_p$: Population density.
$P_{pen}$: Probability of penetration, rotor UAV of 0.25 [13,24].
$P_{mit}$: Probability of mitigation preventing ground fatality, rotor UAV of 0.75 [24].

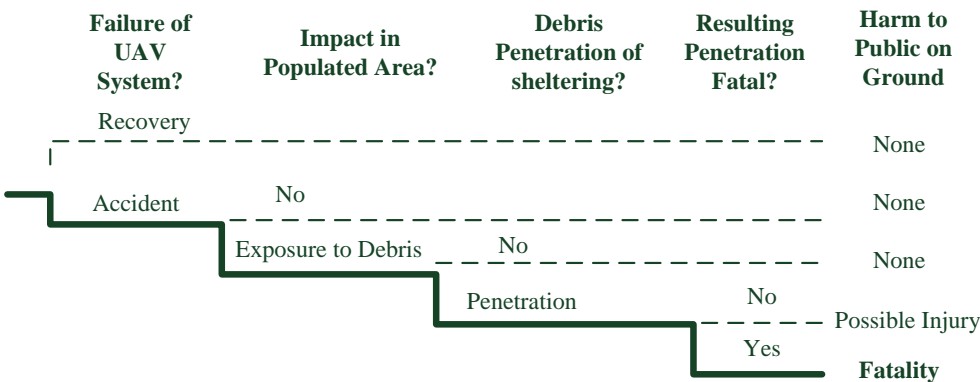

**Figure 12.** Ground impact model [21].

Based on Equation (4), the ELS of outcome in this study is shown as Table 2.

In Table 2, the ELS is relatively high in areas with large populations. However, according to the frontal impact area of different UAVs [11], the smaller the UAV, the lower the risk. In this study, the unsafety rank of ELS for Table 2 are Queiren, Santiman, and FAA area.

**Table 1.** Comparison of risk level for this study and using FAA data [1].

| Test Area | UAV | $P_{event}$ | Population Density (n/m²) | MTBF (h) | $S_{uav}$ (m²) | EF | $P_l$ (%) | KE (J) | M (kg) | V (m/s) |
|---|---|---|---|---|---|---|---|---|---|---|
| Urban, FAA | sUAS | $4.68 \times 10^{-8}$ | 0.0039 | 100 | 0.02 | 0.2 | 0.3 | 82.47 | 0.25 | 25.7 |
| Santiman, PT | Hexa-rotor | $5.37 \times 10^{-8}$ | 0.00013 | 100 | 0.6889 | 0.2 | 0.3 | 384 | 12 | 8 |
| Queiren, TN | Hexa-rotor | $2.69 \times 10^{-7}$ | 0.000651 | 100 | 0.6889 | 0.2 | 0.3 | 384 | 12 | 8 |

**Table 2.** Comparison of the expected level of safety (ELS) for this study with the ground impact model.

| Test Area | UAV | Population Density (n/m²) | 1/MTBF (h) | $S_{uav}$ (m²) | $P_{pen}$ | $(1 - P_{mit})$ | ELS |
|---|---|---|---|---|---|---|---|
| Urban, FAA | sUAS | 0.0039 | 0.01 | 0.02 | 0.25 | 0.25 | $4.875 \times 10^{-8}$ |
| Santiman, PT | Hexa-rotor | 0.00013 | 0.01 | 0.69 | 0.25 | 0.25 | $5.597 \times 10^{-8}$ |
| Queiren, TN | Hexa-rotor | 0.000651 | 0.01 | 0.69 | 0.25 | 0.25 | $2.803 \times 10^{-7}$ |

*4.3. Risk Prevention Tool for the UTM System*

The UAV flight operation must be real time monitoring in the in-flight phase, and analysis in the post-flight phase via the UTM system, as shown in Figure 5. For this reason, this study selects the ADS-B like communication infrastructure in the developing UTM by introducing 4G/LTE and LoRa for flight test 1 and 2, respectively, for sUAs [10,29]. During the in-flight phase, UTM provides the UAV surveillance function using ADS-B like OBU. Unlike manned aircraft, the sUAS are not detectable via radar or other independent surveillance techniques. ADS-B like surveillance on UTM shall be feasible to develop [17]. Mobile communication is the most affordable communication system to adopt, for its wide area deployment. The 4G/LTE cell phone is most available to adopt into UAVs either using cell phones or modules. However, 4G/LTE will not guarantee to cover as high as 400 feet. The developing hierarchical UTM [10] system with the ADS-B like communication adopts devices [21] in high reliability, light weight, low cost, and wide coverage through gateway deployment. Other than 4G/LTE, the LoRa and other proposed technology require constructing and GTS deployment to relay radio surveillance from UAVs into the UTM cloud [10,30,31]. The proposed ADS-B like GTSs receive all UAV surveillance data into the UTM cloud, and distributes to the regional UTM (RUTM) for local governments.

In the UTM operation, DAA is another key function to build. Using software manipulation, DAA can effectively intervene to the controller-pilot communication (CPC) for pilot manipulation to avoidance [30]. With big data collection, UTM FOQA would definitely offer a great contribution to UAV/UAS risk assessment in the future. In the UTM, the real time communication delay by a few seconds, as shown in Table 3, from flight experiences using ADS-B like surveillance infrastructure should be paid attention to. These data will be key factors to affect the UAS risk assessment.

**Table 3.** Real time delay in UTM.

| ADS-B Like | Period | Tx/Rx | Cloud | UTM | C-P |
|:---:|:---:|:---:|:---:|:---:|:---:|
| 4G/LTE | 6~8 | 0.8 | 0.8~1 | 1~2 | 6~10 |
| LoRa | 6~10 | 1~2 | 1~2 | 1~2 | 6~10 |
| APRS | 5~13 | 4~8 | 2~4 | 1~2 | 6~10 |
| Xbee | 6~10 | 1~2 | 1~2 | 1~2 | 6~10 |
| | in seconds | | | | |

## 5. Conclusions

This paper demonstrates an UAV risk assessment using the case study on logistic delivery in remote and suburban areas. Referring to the FAA TCL certification [27], the population density is a major concern to approve the UAS into flight services. The case studies release UAVs in the BVLOS flight. By way of careful path planning, the UAV services can be feasible to meet TCL 3 to reach the accepted risk level to the public.

In this paper, the UAS flight surveillance is effectively monitored under UTM using the ADS-B like infrastructure operating BVLOS under UTM surveillance. Google routing is used for path planning to keep away from the highly populated areas and NFZ. A restricted area and geo-fence is also created to fulfill the UTM requirement for UAVs flying below 400 feet. The ADS-B like infrastructure using 4G/LTE or LoRa is adopted for surveillance with excellent performance. In terms of the risk impact of UAV services to the public, the risk assessment calculates the risk level for a UAV logistic delivery service. The results accomplish very high confidence in the UAV logistic delivery flights with effective data surveillance on UTM.

The risk assessment uses the FAA task force's recommendation [1] for simulations. In terms of actual UAV operations and flight scenarios, the risk level is higher ($10^{-8}$–$10^{-7}$ E/h) than that in the manned air transportation system. However, this result is still acceptable for UAV delivery

application in remote areas and in suburban areas. The real operation of an UAV parcel delivery service in different areas is verified to be efficient and feasible for safe operation.

From the flight tests, the UTM system can completely monitor the logistic delivery flights. It is confident for risk assessment with effective and transparent flight surveillance. The demonstrations use 4G/LTE and LoRa for UTM surveillance. The ADS-B like infrastructure would be more dependent and reliable in surveillance coverage [10,29,30]. Referring to the GTS deployment in Tainan City of Figure 3, the developing ADS-B like infrastructure can offer a seamless UTM surveillance, which mitigates the risk of failure during flight operations.

The safety level assessment using the database for manned aircraft is not realistic because the velocity of a VTOL sUAV is set at 25.7 m/s [1]. For low altitude flights, the feasible velocity lies within 8 m/s for the multiple rotor UAVs.

In conclusion, the risk assessment for UAS using a case study of logistic deliveries in remote and suburban areas demonstrates an acceptable measure for the expected level of safety (ELS). It is feasible to use the UAS routine services for logistic delivery services in Taiwan. This paper merits flight experiments to calculate the expected level of safety using UAVs for logistic delivery under the UTM environment. The result supports the applications of UAV logistic delivery in Taiwan.

**Funding:** This work is financially supported by the Ministry of Science and Technology under contract no. MOST109-2622-E-309-001-CC1.

**Acknowledgments:** The flight tests are conducted by the UTM team in CJCU by Chin E. Lin to support this paper.

**Conflicts of Interest:** There is no conflict of interest to any institutes or individuals, since this is an academic research program.

## Nomenclature

| | |
|---|---|
| $a_s$ | Acceleration including Fall Drag |
| $\rho$ | Air density at sea level (kg/m$^3$) |
| H | Cruise Height (m) |
| V | Cruise Velocity, V = $V_0$ |
| $V_0$ | Cruise Velocity or Initial Velocity (m/s) |
| $C_d$ | Drag coefficient |
| g | Gravitational acceleration (m/s$^2$) |
| $M$ | Maximum Take-off Weight (MTOW) or Mass (kg) |
| $y_0$ | Initial Altitude (m), $y_0$ = H |
| $V_0$ | Initial velocity (m/s) |
| $V_t$ | Terminal velocity (m/s) |
| FIA | Frontal Impact Area (m$^2$) |
| KE | Kinetic Energy (kg m$^2$/s$^2$) |

## Abbreviations

| | |
|---|---|
| 3D | Dirty, Danger, and Dull |
| 4G/LTE | 4th Generation Mobile Communication Long Term Evolution |
| AC | Advisory Circular |
| ADS-B Like | Automatic Dependent Surveillance–Broadcast Like |
| AGL | Above Ground Level |
| APRS | Automatic Packet Reporting System |
| ARC | Air Risk Classifications |
| ATC/ATM | Air Traffic Control/Air Traffic Management |
| BVLOS | Beyond Visual Line of Sight |
| CJCU | Chang Jung Christian University |
| CPC | Controller-Pilot Communication |
| CPD | Closest Point Detection |

| | |
|---|---|
| DAA/SAA | Detect (Sense) and Avoid |
| DoF | Degrees-of-Freedom |
| ELOS | Equivalent Level of Safety |
| ELS | Expected Level of Safety |
| ESC | Electronic Speed Control (Converter) |
| FAA | Federal Aviation Administration, USA |
| FHA | Functional Hazard Assessment |
| FSM | Function Safety Management |
| GA | General Aviation |
| GTS | Ground Transceiver Station |
| HTOL | Horizontal Take-off and Landing (Fixed Wings) |
| ICAO | International Civil Aviation Organization |
| IPP | Integrated Pilot Program |
| ISO | International Standard Organization |
| JAA | Joint Aviation Authority, Europe |
| JARUS | Joint Authorities for Rulemaking of Unmanned System |
| KE | Kinetic Energy |
| LiPo | Lithium Polymer Battery |
| LoRa | Long Range Wide Area Network |
| MAC | Mid-Air Collision |
| MEMS | Microelectromechanical Sensors |
| MRO | Maintenance, Repair and Overhaul |
| MTBF | Mean Time between Failures |
| MTOW | Maximum Take-off Weight |
| NAS | National Airspace System |
| NIA | Non-Integrated Airspace |
| OBU | On-Board Unit |
| RA | Resolution Advisory |
| RUTM/NUTM | Regional UTM/National UTM |
| SORA | Specific Operation Risk Assessment |
| TA | Traffic Advisory |
| TCAS | Traffic Alert and Collision Avoidance System |
| TCL | Technical Capability Level |
| TLS | Target Level of Safety |
| TTC | Time to Conflict |
| UAV | Unmanned Aerial Vehicle |
| UAS | Unmanned Aircraft System |
| UTM | UAS Traffic Management |
| VTOL | Vertical Take-off and Landing (Rotor Wings) |

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
