# Peer review of "Risk Assessment for UAS Logistic Delivery under UAS Traffic Management Environment"

_aerospace, doi:10.3390/aerospace7100140_

Round 1

Reviewer 1 Report

The experiments are still not enough for significant finding(s). Please consider creating statistical samples by repeating those experiments to have some statical measures to support the validity of the study's conclusions. 

Author Response

Thank you for the advice.

Based on Figure 5 for the concept of integrated UAV risk prevention system, this study carries flight tests according to the flight procedures and risk assessment in reference to prevent risk. The validity of the risk assessment process can be established by academic papers, civil aviation organization regulations, such as ICAO, SORA, and FAA.

Due to limited research funding, the empirical experiments were not enough for risk level identification. In future research, the UAV flight procedures will be collected more data to improve analytical basis and quantitative statics for risk assessment.

Reviewer 2 Report

Revised paper is fine. Still a few minor grammar errors. Recommend publication

Author Response

Thank you for the advice. This paper has made improvements.

Reviewer 3 Report

The manuscript entitled "Risk Assessment for UAS Logistic Delivery under UAS Traffic Management Environment", by Pei-Chi Shao studies the ground and air risks for UAS in logistic delivery services. For that purpose, two case studies are performed in Taiwan area.

This reviewer has thoroughly reviewed this manuscript, which has a correct and clear structure. English grammar and style are adequate too.

The topic presented by the authors is very interesting and suitable to be published in this journal. Despite the scientific contents displayed throughout the manuscript are sound, some minor issues were detected:

Section 5 should be renamed as "Conclusion", as discussion is already done in Section 4.

The draft manuscript has no line numbers, so it is difficult to specify the location of the commented issues. A section (S) + paragraph (P) notation has been used:

  • S1 P2: Please, delete "Circular 328-AN/190" and add a complete reference for this document.
  • S1 P3: Delete "and" before "etc." in the last line.
  • S2.1: Use SI accepted abbreviations for hour and second (h, s). Do not use Joules, but J. Numbers under 10 should be spelled out (one, three...). Add a space before some measurement units (e.g., 25m/sec)
  • Figure 3. Delete chinese Characters in the map, or translate into English words.
  • Figure 4. Improve images size, add (a) and (b) to the caption

Author Response

Thank you for the advice.

  1. The section 5 is renamed as Conclusion.
  2. S1P2 suggestion: the study is revised to “document 328 for Unmanned Aircraft Systems (UAS)”.
  3. S1P3 suggestion: this study is deleted “and” before “etc” in the last line.
  4. 1 suggestion: this study is revised all mistakes which are related to the abbreviations of all units in this article.
  5. The area names on the Figure 3. have been added to English. This figure is depicted from Official Earth Planning project to get full map information. It is difficult to find an official map using English. I tried to cover the Chinese characters, but it makes chaotic for reading.

This manuscript is a resubmission of an earlier submission. The following is a list of the peer review reports and author responses from that submission.

Round 1

Reviewer 1 Report

Experiments are not sufficient. Three flight tests for three different scenarios do not prove any significant finding. 

Relying much on results of different other studies and adopting their study cases without verification or justification. 

Some minor editing issues. Ex: line 201/202.. the sentence 'In flight process, UAS Provider should set up standard operation procedures (SOP) to carry pre-flight, in-flight and post-flight' is not understood. 

Reviewer 2 Report

The overall approach of coupling UTM with ADS-B, pilot fllight data and UAS air vehicle design and performance features is very good and practical.

However, the title has "Risk Assessment for UAS" in it, but doesn't review other current UAS Risk Assessment  efforts, such as JARUS Special Operational Risk Assessment (SORA). In regards to our previous UAS Risk Assessment work it only includes the ground risk assessment approach by Melnyk et al in Reference 10. In another paper we address air to air risk and lost of link. It is in "Sense and Avoid Requirements for Unmanned Aircraft Systems Using a Target Level of Safety Approach", 0272-4332/14/0100-0001 014 Society for Risk Analysis. In addition, I have taught a course on Safety By Design and Flight Certification for a number of years.  In 2016, I had a student team conduct a study on "Certification of Small VTOL UAS for Package Delivery" which may be helpful. It is summarized in  "Integration of Functional Safety Management and Systems Engineering for Development Assurance",Engineering, 2017, *, - http://www.scirp.org/journal/eng ISSN Online: 1947-394X, ISSN Print: 1947-3931. It illustrates how a triplex redundant control system with a sense and avoid sensor could meet the TLS and eliminate most of the single point failures. It also recommended the use of ADS-B as a new technology.

The use of MTBFs for all the components only addresses random failures of mechanical components. A Development Assurance (DA) can also address the systematic failures due to human and software errors.

The recent release by EASA of "Proposes Risk-Based Airworthiness Standards for Light Unmanned Aircraft", July 21, 2020 should also be reviewed.

There are some typos and incomplete sentences which need to be corrected.

In summary, I believe the paper is a worthwhile publication illustrating how a practical risk assessment can be made for a coupled UTM and UAS using new technologies.

Reviewer 3 Report

In this research paper, the author demonstrated application of an FAA risk assessment for a UAS delivery case study.  Background information, literature review, and organization document was problematic.  Readability and comprehension of the author's premise were compromised by grammatical errors.  Additionally, the introduction, literature review, and other relevant background components of the paper lacked a cohesive focus.  The author should consider providing focused background information, present the nature of the problem, and clearly articulate the purpose of the research in the first several paragraphs.  The authors suggest the use of "an ADS-B Like" device for UAS tracking and surveillance.  It was unclear to the reviewer how such a device would be differentiated from current ADS-B technology.  The authors employed a case study of a hypothetical delivery.  It was unclear to the reviewer how the case study was selected, the validity of the case study assumptions, and its applicability or generalizability to other UAS delivery flights.  While the risk assessment application was compelling, no new knowledge is presented.